# Predictors for the Occurrence of Seizures in Meningioma

**DOI:** 10.3390/cancers16173046

**Published:** 2024-08-31

**Authors:** Johannes Naegeli, Caroline Sander, Johannes Wach, Erdem Güresir, Jürgen Meixensberger, Felix Arlt

**Affiliations:** Department of Neurosurgery, University Hospital Leipzig, 04103 Leipzig, Germany; johannes.naegeli@t-online.de (J.N.); caroline.sander@medizin.uni-leipzig.de (C.S.); johannes.wach@medizin.uni-leipzig.de (J.W.); erdem.gueresir@medizin.uni-leipzig.de (E.G.); juergen.meixensberger@medizin.uni-leipzig.de (J.M.)

**Keywords:** meningioma, seizures, epilepsy, predictors, surgery

## Abstract

**Simple Summary:**

Seizures are one of the most common and severe symptoms of meningioma, leading to increased morbidity and mortality in the affected patients. Therefore, seizure prevention represents an important goal in the treatment of meningioma patients. For this purpose, our study aims to identify predictors for the occurrence of preoperative and postoperative seizures in meningioma. Neurosurgical tumor resection was demonstrated as an effective treatment of seizures in meningioma patients but is also associated with a moderate risk of new-onset seizures after surgery. The present study identified several independent predictors for seizures in meningioma that could contribute to improved seizure treatment and a deeper understanding of the occurrence of seizures in meningioma patients.

**Abstract:**

Seizure is a common symptom of meningioma that has a major impact on patients’ quality of life. The purpose of this study was to identify predictive factors for the occurrence of preoperative and postoperative seizures. The data of patients with resection of histologically confirmed meningioma at University Hospital Leipzig from 2009 to 2018 were retrospectively examined. Univariate and multivariate logistic regression analyses of different factors influencing seizure outcome were performed. The male gender was identified as an independent positive predictor for preoperative seizures (odds ratio [OR] 1.917 [95% confidence interval {CI} 1.044–3.521], *p* = 0.036), whereas headache (OR 0.230 [95% CI 0.091–0.582], *p* = 0.002) and neurological deficits (OR 0.223; [95% CI 0.121–0.410], *p* < 0.001) were demonstrated to be negative predictive factors. Sensorimotor deficit after surgery (OR 4.490 [95% CI 2.231–9.037], *p* < 0.001) was found to be a positive predictor for the occurrence of postoperative seizures. The identified predictors for the occurrence of seizures in meningioma can contribute to improving seizure treatment and patients’ quality of life.

## 1. Introduction

Meningiomas represent approximately 40% of the neoplasms of the central nervous system and are the most common intracranial tumors [1]. This type of primary tumor has a favorable prognosis due to its benign origin from the arachnoid layer, slow progression and low-grade classification [2,3]. In this context, seizure is one of the most common symptoms in meningioma patients, with multiple contributing factors including genetic predisposition, altered peritumoral microenvironment and different tumor characteristics [4,5]. As a major cause of morbidity and mortality, seizures lead to a significant decrease in quality of life in terms of impaired cognitive function, unemployment and depression [6,7,8]. In addition, more than half of patients with epilepsy suffer from further comorbidities and premature mortality is increased by seizure-related causes of death such as accidents, drownings and suicide [5,9].

For these reasons, seizure prevention represents an important outcome goal in the treatment of meningioma patients. First-line therapy of meningioma, and therefore the control of attributable seizures, consists of neurosurgical resection of the tumor and radiotherapy for atypical and anaplastic tumor grade, recurrence or inaccessible tumor sections [10,11]. The efficacy of prophylactic antiseizure medication (ASM) in patients with brain tumors for the prevention of postoperative seizures remains controversial [12,13,14,15]. In addition, the potential benefits of antiseizure prophylaxis for seizure control must be considered against the adverse effects of antiseizure medications [8,14,15,16]. The current European Association of Neuro-Oncology guidelines do not recommend the prescription of antiseizure medication to reduce the risk of postoperative seizures in patients with brain tumors undergoing surgery [17].

The present study aims to identify independent predictors for the occurrence of seizures in meningioma patients by examining different preoperative and postoperative factors.

## 2. Materials and Methods

### 2.1. Patient Selection

A retrospective monocenter study was performed on patients who underwent microsurgical resection of meningioma at the department of neurosurgery of Leipzig University Hospital between 2009 and 2018. Patient data were identified and obtained from the institutional database and a twelve-month follow-up was observed. Criteria for inclusion were (1) age greater than 18 years, (2) detailed documentation of the analyzed variables in medical records and (3) therapeutic resection of a newly diagnosed and histologically confirmed meningioma.

### 2.2. Variables

Patient data were collected from clinical records of the pathology, radiology and surgical archive systems via the institutional patient management program. The occurrence of preoperative and postoperative seizures was assessed at the first clinical presentation and during the follow-up period. Given that no antiseizure prophylaxis is administered at our department of neurosurgery, and that all patients experiencing preoperative seizures received therapeutic ASM treatment, we did not investigate prophylactic or therapeutic ASM use as factors in our study population. Clinical data including age and gender, comorbidities, headache and neurological deficits attributable to a focal cerebral lesion were collected. The comorbidities studied were hypertension, obesity, heart disease, lung disease, liver disease, kidney disease and diabetes mellitus type 2. Tumor characteristics such as localization, size (maximum diameter <4 cm or ≥4 cm defined according to previous studies [18]), presence of peritumoral edema before and after resection, midline shift and cerebrospinal fluid disorder were acquired from reports of the radiology database. Tumors were classified as skull base tumors, including olfactory groove, suprasellar, sphenoid wing, middle fossa, petrous face, clivus, cerebellopontine angle, or tentorial meningioma, and non-skull base tumors, including falcine, parasagittal, intraventricular and convexity meningioma. In addition, the World Health Organization (WHO) grade according to the definition valid at the time of surgery from pathology reports and the Simpson grade as data on the extent of resection from operation reports were assessed. The preoperative Karnofsky Performance Score (KPS) was recorded at the time of hospital admission (<80 or ≥80, according to previous studies [19]). Duration of surgery, transfusion and surgical complications, including central nervous system (CNS) or wound infection, hydrocephalus and re-craniotomy due to swelling or bleeding, were obtained from medical records. Furthermore, sensorimotor deficits after surgery, adjuvant irradiation and tumor progression, defined as an increase in the diameter of tumor lesions on radiological imaging during the follow-up period, were examined.

### 2.3. Modified STAMPE2 Score

The STAMPE2 score was developed to identify meningioma patients at high risk for postoperative seizures and to guide antiseizure treatment [20]. The score is composed of the following seven factors: sensorimotor deficit, tumor progression, age less than 55 years, major surgical complications including hydrocephalus, re-craniotomy for any reason, CNS infection, preoperative seizure, epileptiform potentials in postoperative electroencephalography (EEG) and preoperative peritumoral edema.

Since postoperative EEG is not routinely performed at our department of neurosurgery, the factor “epileptic potentials in postoperative EEG” was not assessed in our entire study population. To ensure the applicability of the STAMPE2 score to our cohort, a modified version of the score was created excluding the factor “epileptiform potentials in the postoperative EEG”. The modified STAMPE2 score is composed of one point each for sensorimotor deficits, tumor progression, age of less than 55 years and peritumoral edema, plus two points each for major surgical complications and preoperative epilepsy (Table 1).

### 2.4. Statistical Methods

Statistical analysis was performed using IBM SPSS Statistics 29.0 software (IBM, Armonk, NY, USA), MedCalc (version 23.0.1) and Epitools. Continuous variables were expressed as medians with standard deviations and categorical variables as counts and frequencies. Receiver operating characteristic (ROC) analysis was used to identify the optimal cut-off values for the parameters age at tumor resection, duration of surgery and modified STAMPE2 score based on the Youden’s index. Binary univariate regression analysis was conducted to identify predictive factors for the occurrence of preoperative and postoperative seizures. Univariate parameters with *p*-value <0.05 were included in multivariate logistic regression analysis using the enter method to identify independent predictors for preoperative and postoperative seizures in meningioma patients. The statistical significance level for the entire logistic regression was determined as a two-sided *p*-value <0.05. The results were calculated as *p*-values and odds ratios (OR) with 95% confidence intervals (CIs) to assess the factor’s effect size. The chi-square test was used to determine significant differences in the occurrence of postoperative seizures between patients stratified by different modified STAMPE2 score values. The internal validation of the identified predictors for the occurrence of preoperative and postoperative seizures in logistic regression analysis was assessed by bootstrapping with 2.000 resamples. The statistical analysis process is shown in Appendix A.

## 3. Results

### 3.1. Study Population

A total of 396 patients undergoing neurosurgical resection of histologically confirmed meningioma were included in our study. The female to male ratio was 2.2 (273:123) and the median age was 61 years (ranging from 18 to 89 years). Histopathological tumor classification according to the WHO definition revealed 342 WHO grade 1 (86.4%), 49 WHO grade 2 (12.4%) and 5 WHO grade 3 (1.3%) meningiomas. The most frequent tumor location was convexity (n = 94; 23.7%) and the most common tumor type was the meningothelial subtype (n = 160; 40.4%). Demographic and clinical data of the included patients are summarized in Table 2 and Appendix A.

Preoperative seizures were experienced by 64 of the 396 patients (16.2%). After neurosurgical resection, 52 patients (81.2%) were seizure-free, while 12 patients (18.8%) continued to have seizures. Of the 332 study participants (83.8%) without preoperative seizures, 301 (90.7%) remained seizure-free postoperatively, whereas 31 (9.3%) patients experienced new-onset seizures.

### 3.2. Predictors for the Occurrence of Preoperative Seizure

The predictive factors investigated for the occurrence of preoperative seizure are shown in Table 3.

In univariate analysis, preoperative seizures occurred more frequently in the male gender (*p* < 0.001) and non-skull base tumors (*p* = 0.001). In contrast, the WHO grade (*p* = 0.666) and patient age of less than 61 years (*p* = 0.338, {Sens: 54.7; Spec: 52.4}, Appendix A) were not predictors for seizures before surgery in our cohort. An increased rate of preoperative seizures was found in patients with peritumoral edema (*p* = 0.010) and midline shift (*p* = 0.011), whereas tumor size (*p* = 0.108) was not revealed to be a predictive factor. Furthermore, the presence of headache (*p* = 0.002) and neurological deficits (*p* < 0.001) showed a reduced incidence of seizures before surgery. After multivariate logistic regression analysis, male gender (OR 1.917 [95% CI 1.044–3.521], *p* = 0.036), headache (OR 0.230 [95% CI 0.091–0.582], *p* = 0.002) and neurological deficits (OR 0.223; [95% CI 0.121–0.410], *p* < 0.001) remained as independent predictors for the occurrence of preoperative seizures.

### 3.3. Predictors for the Occurrence of Postoperative Seizure

The predictive factors investigated for the occurrence of postoperative seizures are shown in Table 4.

Patient characteristics such as male gender (*p* = 0.023), non-skull base tumors (*p* = 0.030) and preoperative seizures (*p* = 0.030) showed an increased postoperative seizure rate. In contrast, age of more than 68 years (*p* = 0.153, {Sens: 44.2; Spec: 66.9}, Appendix A), preoperative KPS ≥ 80 (*p* = 0.832), extent of tumor resection regarding Simpson grade (*p* = 0.538), tumor progression (*p* = 0.422) and adjuvant irradiation (*p* = 0.446) were not found to be predictors for postoperative seizures. Sensorimotor deficit after surgery (*p* < 0.001) showed an increased incidence of postoperative seizure. However, surgical complications (*p* = 0.490), surgery duration of more than 273 min (*p* = 0.424, {Sens: 53.5; Spec: 53.0}, Appendix A) and postoperative edema (*p* = 0.129) were not predictive factors for seizures after tumor resection. After multivariate logistic regression analysis, sensorimotor deficit (OR 4.490 [95% CI 2.231–9.037], *p* < 0.001) remained as an independent predictor for the occurrence of postoperative seizures.

### 3.4. Application of the Modified STAMPE2 Score

The distribution of the modified STAMPE2 score in our study population is shown in Figure 1 and Table 5.

In our cohort, patients achieved a maximum modified STAMPE2 score of six out of eight points. The chi-square test demonstrated a statistically significant difference in the occurrence of postoperative seizures between patients stratified by modified STAMPE2 score ≥3 (*p* = 0.007) and modified STAMPE2 score ≥4 (*p* ≤ 0.001) (Appendix A). The optimal cut-off value for the modified STAMPE2 score in our study population was identified via ROC analysis for a score ≥4 ({Sens: 30.2; Spec: 90.7}, Appendix A). A total of 30.2% of patients with postoperative seizures are correctly identified by a modified STAMPE2 score of four or more points, whereas 9.3% of patients with no postoperative seizures are incorrectly classified as high-risk patients for postoperative seizures.

## 4. Discussion

The pathogenesis of seizures in patients with meningiomas and brain tumors is not well understood, and multiple factors are assumed to contribute to epileptogenicity. Several pathophysiological mechanisms, including genetic predisposition, inflammation, altered peritumoral and tumoral microenvironment, as well as different tumor characteristics such as type and localization, lead to disturbed neuronal connectivity and therefore to the development of tumor-related seizures [4,5]. As one of the most common symptoms of meningioma, seizures are known to affect patients’ quality of life, leading to increased morbidity and mortality [6,7,8]. For this reason, the identification of predictors for the occurrence of seizures is required to enhance the prevention and treatment of seizures in meningioma patients.

### 4.1. Seizure Rate

In our cohort, 16.2% of participants experienced preoperative seizures, which was comparable to the results of Skardelly et al. (15%) and Zheng et al. (15.2%) but lower than those of Wirsching et al. (31.3%) and Lieu and Howng (28.4%) [20,21,22,23]. After tumor resection, 81.2% of the patients with preoperative seizures achieved seizure freedom, which was higher than in the meta-analysis by Englot et al. (69.3%) [24]. In contrast, new-onset seizures occurred in 7.8% of patients after surgery. These results suggest that tumor resection in meningioma patients with preoperative seizures leads to efficient postoperative seizure control, but it is also associated with a moderate risk of new-onset seizures. Overall, 10.9% of all patients experienced a postoperative seizure, which is lower than Lu et al. (15%) and Islim et al. (17%) [25,26]. The different incidences of seizures could be explained by differences in the size and composition of the study populations as well as different treatment procedures in terms of control intervals and management.

### 4.2. Predictors for Preoperative Seizures

The presence of headache in meningioma patients was associated with a lower rate of preoperative seizure. Headache is known to be one of the most common symptoms in meningiomas, along with seizures and sensorimotor deficits, and has been discussed as less epileptogenic in previous studies [27,28,29]. The assumption that the cause of the headache is related to greater mass effect and the tumor size could not be confirmed in our cohort. In our study population, patients without headaches were four times more likely to have preoperative seizures than patients with headaches. The presentation of headaches may lead to earlier imaging, diagnosis and treatment of meningioma before seizures occur [29].

While meningiomas are more common in women than in men, our study and previous investigations found that preoperative seizures in meningioma patients occur significantly more often in men than in women [1,21,24]. This finding is consistent with the fact that the incidence of epilepsy is generally higher in men than in women [30]. In our study population, male patients were over two times more likely to have preoperative seizures than women. As shown for seizures in general, gender differences in tumor characteristics influenced by neuronal and hormonal receptor status, transcription factors and enzyme expression could be responsible for the observed association [31,32].

Several studies reported an increased incidence of preoperative seizures in the presence of peritumoral edema, which was also a univariate predictor in our cohort [33,34,35]. Peritumoral edema is thought to be related to compression of the adjacent brain parenchyma by the tumor resulting in impaired vascularization, chronic hypoxia, and release of vasogenic factors which increase neuronal excitability and, hence, the potential for seizure development [24,36]. Furthermore, preoperative seizures occurred univariately more often when a midline shift of the brain was detected on radiological imaging, whereas tumor size was not found to be a predictive factor.

Patients with neurological deficits attributable to meningioma were over four times less likely to experience preoperative seizures. A similar association between neurological deficits and preoperative seizures has been reported in previous studies [26,33,35]. As with headaches, the reason could be earlier diagnosis and therefore treatment of meningioma before abnormal activity of the cortical neurons occurs.

### 4.3. Predictors for Postoperative Seizures

Previous studies have shown that patients with preoperative seizures also suffered more frequently from postoperative seizures [25,28]. Therefore, it appears that a low seizure threshold exists even after tumor resection since the altered peritumoral microenvironment persists and remodeling of the disordered neuronal connectivity is in progress. In addition, it was reported that microscopically detected brain invasion of meningioma was over two times more frequently associated with preoperative seizures [37]. Nevertheless, most patients with preoperative seizures benefit from meningioma resection—81.2% of the patients in our cohort became seizure-free. Besides, preoperative seizure did not remain as an independent predictor in the multivariate analysis of our study population.

In accordance with previous studies, patients with sensorimotor deficits after tumor resection were significantly more likely to suffer from postoperative seizures [20,22]. It can be assumed that the manipulation of brain areas to achieve radical resection leads to a disturbance of neuronal structures, resulting in both neurological deficits and an increased propensity for seizures. However, surgical complication in terms of hydrocephalus, re-craniotomy due to swelling or bleeding and CNS infection was not found to be an independent predictor of postoperative seizures in contrast to other studies [25,29,33].

In terms of tumor location, non-skull base meningiomas showed an increased postoperative seizure rate in univariate analysis. Notably, several studies have reported a higher seizure rate and thus hyperexcitability in meningiomas at the convexity adjacent to the motor cortex area than in other cortex areas [26,29,38].

### 4.4. Application of the Modified STAMPE2 Score

Clinical scores for risk assessment of postoperative seizures could improve the prevention and treatment of seizures in meningioma patients. Therefore, Wirsching et al. developed the STAMPE2 score to identify patients at high risk for postoperative seizures and to guide antiseizure treatment in meningioma patients [20]. A modified STAMPE2 score, which excluded the factor “epileptic potentials in postoperative EEG”, was applied to our study population.

The chi-square test and ROC analysis results demonstrated that risk stratification for postoperative seizures in meningiomas is feasible for patients with a modified STAMPE2 score ≥4 versus a modified STAMPE2 score <4 (*p* ≤ 0.001). Therefore, a modified STAMPE2 score of four or more points might be considered for guiding antiseizure treatment for postoperative seizure control in meningioma patients. Due to insufficient evidence, the prescription of antiseizure medication to reduce the risk of postoperative seizures in patients with brain tumors undergoing surgery is not recommended in the current European Association of Neuro-Oncology guidelines (Level C) [17]. In this regard, the multicenter STOP’EM study is currently investigating the potential risk reduction of postoperative seizures after meningioma resection through antiseizure prophylaxis with levetiracetam and its effects on the quality of life of seizure-free meningioma patients.

## 5. Limitations

Due to the retrospective design of our study, data collection was limited to information from patient records, operative reports and varying quality of documentation leading to an overall bias. Since all patients with seizures at our department received only ASM therapy, no conclusion could be provided on the benefit of prophylactic antiseizure treatment. It should also be noted that the statistical analysis may be limited in terms of reliability and validity due to the small size of patient subgroups. The potential confounding effects and interactions between predictors identified in the binary univariate logistic regression analysis were addressed in a subsequent multivariate regression analysis. Based on the twelve-month follow-up period, only limited conclusions can be drawn about the patients’ outcomes in terms of tumor progression and late postoperative seizures. Therefore, further studies are needed to validate the independent factors found in our study population.

## 6. Conclusions

Seizures are one of the most common and severe symptoms in meningioma patients. Therefore, seizure control remains a primary outcome goal in the treatment of patients with meningioma. In our study population, male gender was identified as a positive predictor for preoperative seizures, whereas headache and neurologic deficits were found to be negative predictors. In contrast, sensorimotor deficit after surgery was shown to be a positive predictor for postoperative seizures in meningioma patients. The studied predictors for the occurrence of seizures in meningioma can contribute to improving seizure treatment and patients’ quality of life.

## Figures and Tables

**Figure 1 cancers-16-03046-f001:**
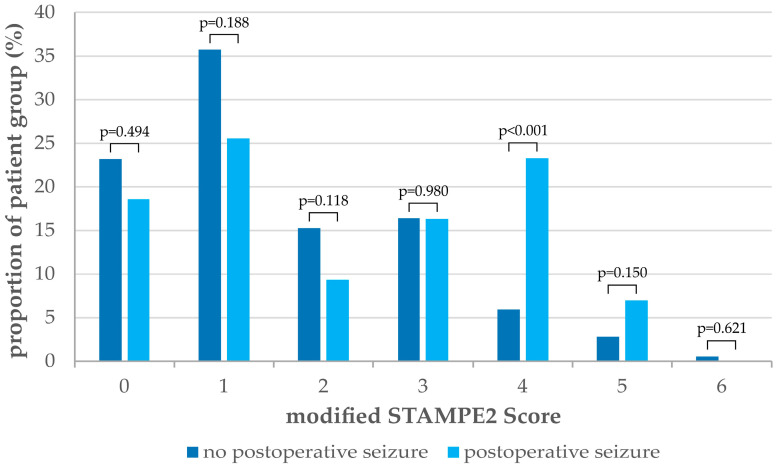
Distribution of the modified STAMPE2 score in patients with postoperative seizures (N = 43) compared to patients with no postoperative seizures (N = 353). Scores of seven and eight were not achieved in the study population.

**Table 1 cancers-16-03046-t001:** Modified STAMPE2 Score.

**S**ensorimotor Deficit	1 Point
**T**umor progression	1 Point
**A**ge < 55 y	1 Point
**M**ajor surgical complication ^1^	2 Point
**P**reoperative epilepsy	2 Point
**E**dema	1 Point

^1^ Hydrocephalus, re-craniotomy for any reason and CNS infection.

**Table 2 cancers-16-03046-t002:** Demographic and clinical patient characteristics.

**Characteristic**	**Preoperative Seizures**	**Postoperative Seizures**	**Preoperative and Postoperative Seizures**	**No Seizures**	**Total Number of Patients**
	N (%)	N (%)	N (%)	N (%)	N (%)
	52 (13.2)	31 (7.8)	12 (3.0)	301 (76.0)	396 (100)
Age at resection, year					
Range	33–81	33–82	40–81	18–89	18–89
Median ± SD	60.00 ± 12.54	68.00 ± 12.86	55.00 ± 13.83	61.00 ± 13.37	61.00 ± 13.27
Sex					
Male	29 (55.8)	17 (54.8)	3 (25.0)	74 (24.6)	123 (31.1)
Female	23 (44.2)	14 (45.2)	9 (75.0)	227 (75.4)	273 (68.9)
WHO grade					
1	46 (88.5)	28 (90.3)	7 (58.4)	261 (86.7)	342 (86.4)
2	6 (11.5)	3 (9.7)	4 (33.3)	36 (12.0)	49 (12.4)
3	0 (0.0)	0 (0.0)	1 (8.3)	4 (1.3)	5 (1.2)
Tumor location					
Non-skull base	31 (59.6)	16 (51.6)	10 (83.3)	120 (39.9)	177 (44.7)
Skull base	21 (40.4)	15 (48.4)	2 (16.7)	181 (60.1)	219 (55.3)
Tumor size, maximal diameter					
<4 cm	26 (50.0)	20 (64.5)	5 (41.7)	177 (58.8)	228 (57.6)
≥4 cm	26 (50.0)	11 (35.5)	7 (58.3)	124 (41.2)	168 (42.4)
Preoperative edema					
Yes	32 (61.5)	14 (45.2)	6 (50.0)	124 (41.2)	176 (44.4)
No	20 (38.5)	17 (54.8)	6 (50.0)	177 (58.8)	220 (55.6)
Midline shift					
Yes	24 (46.2)	8 (25.8)	5 (41.7)	88 (29.2)	125 (31.6)
No	28 (53.8)	23 (74.2)	7 (58.3)	213 (70.8)	271 (68.4)
Simpson grade					
1	17 (32.7)	18 (58.1)	7 (58.3)	156 (51.8)	198 (50.0)
2	27 (51.9)	7 (22.6)	2 (16.7)	84 (27.9)	120 (30.3)
3	3 (5.8)	2 (6.4)	1 (8.3)	16 (5.3)	22 (5.6)
4	5 (9.6)	4 (12.9)	2 (16.7)	45 (15.0)	56 (14.1)
5	0 (0.0)	0 (0.0)	0 (0.0)	0 (0.0)	0 (0.0)
Preoperative neurological deficit					
Yes	19 (36.5)	22 (71.0)	6 (50.0)	227 (75.4)	274 (69.2)
No	33 (63.5)	9 (29.0)	6 (50.0)	74 (24.6)	122 (30.8)
Sensorimotor deficit					
Yes	8 (15.4)	13 (41.9)	5 (41.7)	37 (12.3)	63 (15.9)
No	44 (84.6)	18 (58.1)	7 (58.3)	264 (87.7)	333 (84.1)
Tumor progression					
Yes	3 (5.8)	3 (9.7)	0 (0.0)	12 (4.0)	18 (4.5)
No	49 (94.2)	28 (90.3)	12 (100)	289 (96.0)	378 (95.5)
Preoperative KPS ≥ 80					
Yes	20 (38.5)	17 (54.8)	5 (41.7)	139 (46.2)	181 (45.7)
No	32 (61.5)	14 (45.2)	7 (58.3)	162 (53.8)	215 (54.3)

Abbreviations: SD—standard deviation; WHO—World Health Organization; KPS—Karnofsky Performance Score.

**Table 3 cancers-16-03046-t003:** Predictors for preoperative seizures.

Predictive Factor	Univariate Analysis		Multivariate Analysis	
	OR (95% CI)	*p*-Value	OR (95% CI)	*p*-Value
Age < 61	1.301 (0.759–2.230)	0.338		
Male gender	**2.648 (1.534–4.572)**	**<0.001**	**1.917 (1.044–3.521)**	**0.036**
WHO grade	not applicable	0.666		
Non-skull base	**2.569 (1.474–4.477)**	**0.001**	1.787 (0.958–3.333)	0.068
Tumor size ≥ 4 cm	1.553 (0.908–2.657)	0.108		
Preoperative edema	**2.055 (1.192–3.542)**	**0.010**	1.692 (0.852–3.358)	0.133
Midline shift	**2.037 (1.179–3.518)**	**0.011**	1.765 (0.897–3.470)	0.100
Headache	**0.243 (0.102–0.583)**	**0.002**	**0.230 (0.091–0.582)**	**0.002**
Neurological deficits	**0.214 (0.122–0.374)**	**<0.001**	**0.223 (0.121–0.410)**	**<0.001**

Significant *p*-values (*p* < 0.05) with OR (95% CI) in univariate analysis and multivariate analysis are bold. Abbreviations: OR—odds ratio; CI—confidence interval; WHO—World Health Organization.

**Table 4 cancers-16-03046-t004:** Predictors for postoperative seizures.

Predictive Factor	Univariate Analysis		Multivariate Analysis	
	OR (95% CI)	*p*-Value	OR (95% CI)	*p*-Value
Age ≥ 68 years	1.597 (0.841–3.033)	0.153		
Male gender	**2.111 (1.111–4.009)**	**0.023**	1.863 (0.949–3.657)	0.071
WHO grade	not applicable	0.566		
Non-skull base	**2.046 (1.072–3.906)**	**0.030**	1.546 (0.780–3.065)	0.212
Tumor size ≥ 4 cm	0.974 (0.513–1.851)	0.937		
Preoperative seizures	**2.241 (1.081–4.643)**	**0.030**	1.769 (0.814–3.844)	0.150
Preoperative KPS ≥ 80	0.933 (0.494–1.765)	0.832		
Simpson grade	not applicable	0.538		
Surgical complications ^1^	1.319 (0.601–2.895)	0.490		
CNS infection	0.396 (0.052–3.030)	0.373		
Hydrocephalus	1.657 (0.189–14.525)	0.648		
Re-craniotomy due to swelling/bleeding	1.404 (0.638–3.088)	0.399		
Postoperative edema	1.859 (0.836–4.136)	0.129		
Sensorimotor deficit	**4.928 (2.492–9.745)**	**<0.001**	**4.490 (2.231–9.037)**	**<0.001**
Surgery duration ≥ 273 min	1.295 (0.687–2.444)	0.424		
Adjuvant irradiation	1.543 (0.506–4.710)	0.446		
Tumor progression	1.690 (0.469–6.092)	0.422		

Significant *p*-values (*p* < 0.05) with OR (95% CI) in univariate analysis and multivariate analysis are bold. Abbreviations: OR—odds ratio; CI—confidence interval; WHO—World Health Organization; KPS—Karnofsky Performance Score; CNS—central nervous system. ^1^ Hydrocephalus, re-craniotomy and CNS infection.

**Table 5 cancers-16-03046-t005:** Distribution of the modified STAMPE2 score.

**Modified STAMPE2 Score**	**No Postoperative** **Seizures**	**Postoperative** **Seizures**	**Chi-Square Test**
	N (%)	N (%)	*p*-Value
0	82 (23.2)	8 (18.6)	n.a.
1	126 (35.7)	11 (25.6)	0.624
2	54 (15.3)	4 (9.3)	0.093
3	58 (16.4)	7 (16.3)	**0.007**
4	21 (5.9)	10 (23.3)	**<0.001**
5	10 (2.8)	3 (7.0)	0.461
6	2 (0.6)	0 (0.0)	n.a.
7	0 (0.0)	0 (0.0)	n.a.
8	0 (0.0)	0 (0.0)	n.a.

Significant *p*-values (*p* < 0.05) in the chi-square test for different cut-off values (≥ vs.<) of the modified STAMPE2 score are bold. Abbreviations: n.a.—not applicable.

## Data Availability

The data presented in this study are available upon reasonable request from the corresponding author.

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
