# Peer review of "Predictors for the Occurrence of Seizures in Meningioma"

_cancers, 2024, doi:10.3390/cancers16173046_

Round 1

Reviewer 1 Report (Previous Reviewer 2)

Comments and Suggestions for Authors

Dear Authors,

I read the new version of the article “Predictors for the occurrence of seizures in meningioma” and the responses to the comments made after reading the first version.

I rate the changes made positively. I have no further comments.

Best regards.

Author Response

Thank you for your review and your feedback on our revised manuscript. We are pleased that the revision of our manuscript based on your comments has been positively rated.

Reviewer 2 Report (Previous Reviewer 3)

Comments and Suggestions for Authors

The manuscript by Naegeli et al. is a revised version of the cancers-3036859 manuscript, concerning with possible predictors of preoperative and postoperative seizures related to meningiomas.

In the current version, the authors successfully dismissed a number of questionable statements, concentrating on the primary aim chosen.

While the general storyline of the manuscript was crystallised effectively, some visualisations and clarifications still are required.

[R1]

The figure depicting the pipeline of the strategy for identifying independent predictive factors for preoperative and postoperative seizures in meningioma patients is missing in the main text.

Such visualisation of the pipeline steps performed and the timeline would be of great help for future readers.

[R1a]

Please include the scheme of the pipeline depicting the strategy for identifying independent predictive factors as a separate main Figure, clarifying all steps performed in a step-by-step fashion.

In particular, the following steps must be reflected on the pipeline scheme:

- the dataset and its kea properties

- preliminary univariate regression analysis

- multivariate logistic regression analysis

- validation of the built model via bootstrap

- performance indicators estimated

Some examples of pipelines can be found in the following papers:

The Fig. 1 in
Huang et al. (2022). Cancers, 14(19), 4869. https://doi.org/10.3390/cancers14194869          

The Fig. 1 in
Lin et al. (2023). Cancers, 15(14), 3690. https://doi.org/10.3390/cancers15143690

The Fig. 1 in
Gomez et al. (2022). Cancers, 14(12), 2922. https://doi.org/10.3390/cancers14122922

The Fig. 1 in
Franco et al. (2022). Cancers, 14(14), 3506. https://doi.org/10.3390/cancers14143506

The Fig. 1 in
Mandal et al. (2023). Cancers, 15(5), 1428. https://doi.org/10.3390/cancers15051428

[R1b]

All steps depicted on the pipeline scheme figure must be cross-referenced in the “Statistical methods” subsection, and vice versa.

[R2]

The details of the bootstrap validation strategy used in this study to assess built regression-based models are still missed in the text of the manuscript.

[R2a]

In the relevant subsection of the “Materials and Methods” section, the authors must explain the bootstrap validation strategy applied in full details.

[R2b]

The achieved ROC curves (that had been presented in the previous answer but are currently missed in the supplement) must be depicted as an additional Figure, with corresponding uncertainty bands estimated via bootstrap strategy visualised alongside ROCs.

To ensure visualisations that are clear for readers, please see some examples of well-depicted ROC curves with CI intervals in the following papers:

The Fig. 4 and the Fig 5 in
Zhang et al. (2020). Cancers, 12(3), 622. https://doi.org/10.3390/cancers12030622

The Fig. 30 in
Sarafidis et al. (2022). Cancers, 14(14), 3358. https://doi.org/10.3390/cancers14143358

The Fig. 4 in
Barberis et al. (2021). Metabolites, 11(12), 847. https://doi.org/10.3390/metabo11120847

The Fig.5 in
MacIntyre et al. (2023). Metabolites, 13(10), 1071. https://doi.org/10.3390/metabo13101071

The Fig. 1 in
Sokollik et al. (2023). Diagnostics, 13(15), 2491. https://doi.org/10.3390/diagnostics13152491

The Fig.6 in
Somtua et al. (2023). Metabolites, 13(11), 1135. https://doi.org/10.3390/metabo13111135

[R2c]

The achieved AUC performance indicators (that had been presented in the previous answer but are currently missed in the supplement) must be reported for multivariate models built, with AUC variances estimated via cross-validation or other technique.

[R3]

Please include to the supplement all demography visualisations that had been presented in the previous answer but are currently missed in the supplement.

[R4]

Supplementary materials were not included into the current version of the submission.

Please include all supplementary materials critical for understanding the  results achieved.

[R5]

The strategy for identifying predictive variables via preliminary binary univariate regression analysis is possible yet too restrictive, which might result in missing critical variables that impacted the seizure development non-additively – via interactions with other variables.

Please identify this critical limitation of the strategy for identifying predictive variables within the “Limitation” section of this manuscript.

[R6]

Please consider to add figures with proportions and their CIs gained via online freeware “Epitools” (https://epitools.ausvet.com.au/chisq) into the text of the manuscript or into the supplement.

Note. Due to limited numbers in the score-6 group, this group should be visualised aggregately with the score 5 group.

[R7]

There is no need to hoard the abstract with specific values of performance indicators and CIs. Instead, please stress out the primary results gained.

Some suggestions for the future work (not for this manuscript).

[S1]

In the present version, both STAMPE2 and modSTAMPE2 scales are to discrete, which might result in the poor resolution of the patient groups.

Please consider to modify weights for the components of the STAMPE2 score, assessing the results gained via performance indicators.

[S2]

Please consider to apply to the raw dataset any modern ML-based techniques like Random Forests or the Decision-tree-based boosting (XG boost, catboost, etc) to identify possible non-linear relations between the variables tested, identifying concealed predictor variable if any being missed in the present study.

Comments on the Quality of English Language

Language

[R8]

Some clarifications are required

[LL65-66] “by examining different … factors” – various? Please clarify.

[L101] “<80 or ≥80 according to previous studies” – cutoff criteria? Please clarify.

[L109] “The STAMPE2 score was developed” – In this study? Earlier? Please clarify.

Author Response

Response to Reviewer 3 Comments
1.
Summary
Thank you for your detailed review and valuable feedback on our manuscript.
Supplementary materials have now been included in the revised manuscript to facilitate a more comprehensive understanding of the results obtained in the present study. Please find the detailed responses below, along with the corresponding revisions in track changes in the re-submitted file.
Thank you for the additional suggestions for future work. The application of modern ML-based techniques and the modification of the weights for the components of the STAMPE2 score will be included in future analyses.
2.
Point-by-point response to Comments and Suggestions for Authors
Comment 1:
[R1]
The figure depicting the pipeline of the strategy for identifying independent predictive factors for
preoperative and postoperative seizures in meningioma patients is missing in the main text.
Such visualisation of the pipeline steps performed and the timeline would be of great help for future
readers.
[R1a]
Please include the scheme of the pipeline depicting the strategy for identifying independent
predictive factors as a separate main Figure, clarifying all steps performed in a step-by-step fashion.
In particular, the following steps must be reflected on the pipeline scheme:
- the dataset and its kea properties
- preliminary univariate regression analysis
- multivariate logistic regression analysis
- validation of the built model via bootstrap
- performance indicators estimated
Some examples of pipelines can be found in the following papers:
The Fig. 1 in
2
Huang et al. (2022). Cancers, 14(19), 4869. https://doi.org/10.3390/cancers14194869
The Fig. 1 in
Lin et al. (2023). Cancers, 15(14), 3690. https://doi.org/10.3390/cancers15143690
The Fig. 1 in
Gomez et al. (2022). Cancers, 14(12), 2922. https://doi.org/10.3390/cancers14122922
The Fig. 1 in
Franco et al. (2022). Cancers, 14(14), 3506. https://doi.org/10.3390/cancers14143506
The Fig. 1 in
Mandal et al. (2023). Cancers, 15(5), 1428. https://doi.org/10.3390/cancers15051428
[R1b]
All steps depicted on the pipeline scheme figure must be cross-referenced in the “Statistical methods”
subsection, and vice versa.
Response 1:
A pipeline scheme figure of the strategy for identifying independent predictive factors for preoperative and postoperative seizures in meningioma patients has been included in the revised manuscript. All steps of the statistical analysis process have been described in the “statistical methods” subsection of the manuscript.
Comment 2:
[R2]
The details of the bootstrap validation strategy used in this study to assess built regression-based
models are still missed in the text of the manuscript.
[R2a]
In the relevant subsection of the “Materials and Methods” section, the authors must explain the
bootstrap validation strategy applied in full details.
[R2b]
The achieved ROC curves (that had been presented in the previous answer but are currently missed
in the supplement) must be depicted as an additional Figure, with corresponding uncertainty bands
estimated via bootstrap strategy visualised alongside ROCs.
To ensure visualisations that are clear for readers, please see some examples of well-depicted ROC
curves with CI intervals in the following papers:
The Fig. 4 and the Fig 5 in
Zhang et al. (2020). Cancers, 12(3), 622. https://doi.org/10.3390/cancers12030622
The Fig. 30 in
Sarafidis et al. (2022). Cancers, 14(14), 3358. https://doi.org/10.3390/cancers14143358
The Fig. 4 in
Barberis et al. (2021). Metabolites, 11(12), 847. https://doi.org/10.3390/metabo11120847
The Fig.5 in
3
MacIntyre et al. (2023). Metabolites, 13(10), 1071. https://doi.org/10.3390/metabo13101071
The Fig. 1 in
Sokollik et al. (2023). Diagnostics, 13(15), 2491. https://doi.org/10.3390/diagnostics13152491
The Fig.6 in
Somtua et al. (2023). Metabolites, 13(11), 1135. https://doi.org/10.3390/metabo13111135
[R2c]
The achieved AUC performance indicators (that had been presented in the previous answer but are
currently missed in the supplement) must be reported for multivariate models built, with AUC
variances estimated via cross-validation or other technique.
Response 2:
The bootstrap validation strategy with 2.000 resamples applied to our study cohort has been explained in the “Materials and Methods” section of the revised manuscript.
The ROC curves with CI intervals and AUC performance indicators for the parameters age at tumor resection, surgery duration and modified STAMPE2 score from our previous response have been included as supplementary material.
Comment 3:
[R3]
Please include to the supplement all demography visualisations that had been presented in the
previous answer but are currently missed in the supplement.
Response 3:
Demography visualizations of the parameter age at tumor resection, surgery duration and preoperative KPS have been included in the supplement materials.
Comment 4:
[R4]
Supplementary materials were not included into the current version of the submission.
Please include all supplementary materials critical for understanding the results achieved.
Response 4:
The revised manuscript includes supplementary materials that are essential for understanding of the results achieved in the present study.
Comment 5:
[R5]
The strategy for identifying predictive variables via preliminary binary univariate regression analysis
is possible yet too restrictive, which might result in missing critical variables that impacted the seizure
development non-additively – via interactions with other variables.
Please identify this critical limitation of the strategy for identifying predictive variables within the
“Limitation” section of this manuscript.
Response 5:
In univariate binary logistic regression analysis, potential confounding effects and interactions between the different identified predictors are not addressed, which may result in inaccurate results. This issue has been addressed in the multivariate analysis to correct for such biases. The limitations of this approach have been clearly stated in the manuscript's 'Limitations' section.
Comment 6:
[R6]
Please consider to add figures with proportions and their CIs gained via online freeware “Epitools”
(https://epitools.ausvet.com.au/chisq) into the text of the manuscript or into the supplement.
Note. Due to limited numbers in the score-6 group, this group should be visualised aggregately with
the score 5 group.
Response 6:
Figures showing proportions and 95% confidence intervals of patients with postoperative seizures dichotomized into different modified STAMPE2 cut-off values have been added to the supplementary material.
Comment 7:
[R7]
There is no need to hoard the abstract with specific values of performance indicators and CIs. Instead,
please stress out the primary results gained.
Response 7:
The comprehensive presentation of the identified predictors for the occurrence of seizures in meningioma patients with performance indicators and confidence intervals in the abstract clarifies the central findings of our study. Therefore, we kindly request your understanding regarding our preference to maintain this information in the abstract.
Some suggestions for the future work (not for this manuscript).
[S1]
In the present version, both STAMPE2 and modSTAMPE2 scales are to discrete, which might result
in the poor resolution of the patient groups.
Please consider to modify weights for the components of the STAMPE2 score, assessing the results
gained via performance indicators.
[S2]
Please consider to apply to the raw dataset any modern ML-based techniques like Random Forests
or the Decision-tree-based boosting (XG boost, catboost, etc) to identify possible non-linear relations
between the variables tested, identifying concealed predictor variable if any being missed in the
present study.
Response S1 and S2:
Thank you very much for your favorable suggestions for our future work! The application of modern ML-based techniques to the dataset and the modification of the weights for the components of the STAMPE2 score will be included in our future research. Your recommendations will enhance the clarity and accuracy of our data analysis.

Round 2

Reviewer 2 Report (Previous Reviewer 3)

Comments and Suggestions for Authors

The authors STILL have not addressed some critical points on data uncertainty visualisation.

[R2-1]

The achieved ROC curves must be depicted as an additional Figure 2 in the main text, with corresponding uncertainty bands (SD, CI90, CI95 or another) estimated via bootstrap strategy and visualised alongside ROCs.

To ensure visualisations that are clear for readers, please see some examples of well-depicted ROC curves with uncertainty bands in the following papers:

The Fig. 4 and the Fig 5 in Zhang et al. (2020). Cancers, 12(3), 622. https://doi.org/10.3390/cancers12030622

The Fig. 30 in Sarafidis et al. (2022). Cancers, 14(14), 3358. https://doi.org/10.3390/cancers14143358

The Fig. 4 in Barberis et al. (2021). Metabolites, 11(12), 847. https://doi.org/10.3390/metabo11120847

The Fig.5 in MacIntyre et al. (2023). Metabolites, 13(10), 1071. https://doi.org/10.3390/metabo13101071

The Fig. 1 in Sokollik et al. (2023). Diagnostics, 13(15), 2491. https://doi.org/10.3390/diagnostics13152491

The Fig.6 in Somtua et al. (2023). Metabolites, 13(11), 1135. https://doi.org/10.3390/metabo13111135

Please also note that AUCs equal to 0.5 are not a problem due to simplistic nature of logistic regression, which is not optimal for hard classification tasks.

[R2-2]

The figure 1 still presents the gained results in a suboptimal way and must be extended by an additional panel 1B.

For example, it is possible to depict the table 1 derived data, using CI95 (or CI90) intervals assessed by online freeware “Epitools” (https://epitools.ausvet.com.au/chisq) in the way depicted in the attachment file.

Please add an additional Fig. 1B panel depicting the primary result of the manuscript.

Comments on the Quality of English Language

Please check for remaining typos

Author Response

This manuscript is a resubmission of an earlier submission. The following is a list of the peer review reports and author responses from that submission.

Round 1

Reviewer 1 Report

Comments and Suggestions for Authors

This clinical research study aims to identify the predictors of seizures in patients diagnosed with meningioma. All patients underwent therapeutic resection of a newly diagnosed and histologically confirmed meningioma.  This study provides remarkably valuable and insightful results, making a significant contribution to the field of clinical meningioma management. Predicting seizures and identifying high-risk patients are crucial for establishing a risk-based AED treatment strategy.

I have several remarks.

1. The authors used a modified version of the STAMPE2 score. The STAMPE2 score was proposed by Wirsching et all in 2016 to identify patients at high risk for postoperative seizures. Wirsching et all used STAMPE2 scores to guide anticonvulsant treatment, but the recent paper, first, the modified STAMPE2 scoring system and, second, used the modified system for predicting seizures associated with meningioma. Why did the STAMPE2 scoring system need changes? Did STAMPE2 have limitations that affected its accuracy and usability? The last statement of Abstract: "The STAMPE2 score could not reliably identify patients at high risk for postoperative seizures in our study group." Was the modified STAMPE2 unreliable? Why did not modifications of STAMPE2 improved its reliability?      

2. Preoperative AED use appears to be an overlooked predictive factor. The authors mentioned: "patients experiencing seizures received AED treatment, we did not investigate this factor in our study population".  However, 52 patients (13.2%) showed preoperative seizures. Did these patients receive pharmacotherapy (medication) as part of their treatment? Could AEDs influence meningioma growth and postoperative outcomes in these patients?

Minor comments.

1. Please add missing references "Error! Reference source not found" in Page 3 (line 119) Page 6 (line 165), Page 7 (line 184), Page 8 (line 207, 223)

2. Table 1 is missing from the document, while Table 2 appears to be duplicated.

Reviewer 2 Report

Comments and Suggestions for Authors

Dear Authors,

the manuscript requires a preliminary editorial revision: 1) table 2 is reported twice; 2) in table 5 the numbers reported in the second column (total: 353) do not correspond to patients with preoperative seizures; 3) in the caption of figure 2 “six months” (line 251) seems incorrect; 4) line 270: 9,3% or 7,8%?.

I point out to the authors, but I leave it up to them to choose the term to use, that recent recommendations from the ILAE (International league against epilepsy) encourage the replacement of the term AED (used several times starting from line 56) with ASM (anti seizure medications), since drugs do not change the natural history of epilepsy. Similarly, rather than a prophylactic action of the drugs (line 57), we can speak of their cosmetic effect on seizures.

Best regards.

Reviewer 3 Report

Comments and Suggestions for Authors

The manuscript by Naegeli et al. assesses possible predictors of perioperative seizures related to meningiomas.

Initially, the authors tested the impact of various variables on seizure appearance, using suboptimal ML-based methods and without data pipeline reported. No considerations on possible factor-to-factor interactions were provided in the manuscript.

Then, the predictive value of modified STAMPE2 scores was assessed, using unidentified data pipeline, with too mechanistic interpretations of the results gained. Based on the Table 5 data, it looks like that modified STAMPE2 scores might be sensitive for patients at high risk for postoperative seizures, beginning from 3+ modified STAMPE2 scores. Thus, the authors identified that some factors, which had been included into modified STAMPE2 scale might impact the prediction seriously.

Additionally, the possible impact of seizures and surgical treatment on patient's quality of life was assessed using Karnofsky Performance Score.

The general storyline is clear, despite the suboptimal data pipelines used.

Regardless of this, the interpretations of the gained results are too mechanistic, without considerations on the variability of underlying factors associated with meningioma-related seizures.

The authors must specify data processing pipelines in a clear way to ensure sound interpretations of the results.

Critical issue

[1]

In the “Informed Consent Statement” [LL428-429], the authors claimed that “Patient consent was waived due to the retrospective character of this study and the approval of our Ethics Committee”.

Despite the claim, informed consent is one of the founding principles of research ethics.

According to CIOMS (WHO) guidelines, a REC may approve a modification or waiver of informed consent to research if:

1. the research would not be feasible or practicable to carry out without the waiver or modification;

2. the research has important social value;

and

3. the research poses no more than minimal risks to participants

Due to legal issues, the following actions and changes are critical for this manuscript.

[1a]

For details, please read https://researchsupport.admin.ox.ac.uk/governance/ethics/resources/consent

and other relevant sources to understand legal issues relevant to informed consents.

Please also consult with the Declaration of Helsinki, the GPDR, as well as papers by Dal-Ré 2023; Uedo, Ponchon 2022 and other relevant sources describing the problem of informed consents for Retrospective studies.

[1b]

In the text of the “Informed Consent Statement”, please clearly indicate all reasons to waive informed consents for all subjects, referencing directly all relevant guidelines allowing waiving.

[1c]

According to general ethics guidelines (the Declaration of Helsinki and Personal Data regulations like GPDR), it is highly recommended to gain the written informed consent from every alive participant enrolled to this study.

In case of mentally disabled/deceased participants, it is highly recommended to obtain the “written informed consent” from their Legal representatives / heirs on the prospective use of personal data in this study.

Please consider to fulfil these basic requirements.

For reference, please see

Dal-Ré, R. Waivers of informed consent in research with competent participants and the Declaration of Helsinki. Eur J Clin Pharmacol 79, 575–578 (2023). https://doi.org/10.1007/s00228-023-03472-w

Uedo, N., & Ponchon, T. (2022). Methods to obtain informed consent in medical and biological research involving human subject: application to studies on digestive endoscopy. Endoscopy international open, 10(6), E719–E720. https://doi.org/10.1055/a-1776-7801

Major issues

[2]

The data reporting must be improved extensively.

[2a]

The Fig 2 must be visualised in a form of a Raincloud plot with jittering, the Median, IQR boxes and CI95 intervals depicted for each quantitative characteristic.

For better visualisations, please consider to depict the Fig. 2 as three separate panels comparing “No seizures” vs “Preoperative seizures” groups, “No seizures” vs “Postoperative seizures” groups, and “No seizures” vs “Pre- and postoperative seizures” groups individually.

https://jasp-stats.org/2022/07/29/bayesian-repeated-measures-anova-an-updated-methodology-implemented-in-jasp/

To achieve this, it is possible to use the freeware JASP https://jasp-stats.org/ or another freeware/software capable of drawing Raincloud plots with jittering.

Please also consider to report the Bayesian three-way mix-design ANOVA results (Factors – preoperative, postoperative and time of measurement) in a way similar to the described:

https://jasp-stats.org/2022/07/29/bayesian-repeated-measures-anova-an-updated-methodology-implemented-in-jasp/

Traditional three-way mix-design ANOVA’s results also might be reported.

[2b]

No raw data were provided with the manuscript as supplementary tables or data links, making the results reported to be poorly verifiable.

Please consider to include the raw numerical data gained and the results for all methods applied to the Supplement as XLS or CSV files.

In particular, all details on the individual statistical tests performed by the authors, including dfs and other critical indicators also must be reported.

[3]

The demography visualisations also must be improved.

[3a]

To ensure clear comprehension of the demographic and clinical characteristics, the quantitative indicators like “Age” must be additionally visualized in the Supplement (or near the Table 2 in the main text) as Raincloud plots with jittering, the Median, IQR boxes and CI95 intervals visualized between different groups.

[3b]

Please also consider to report as Raincloud plots with jittering the raw data on quantitative indicators like “Tumor size” (in cms) that were thresholded intentionally.

[3c]

The differences between groups must be assessed statistically, by ANOVA.

To achieve these points of review, it is possible to use the freeware JASP (https://jasp-stats.org/) or code (https://datavizpyr.com/grouped-violinplot-with-jittered-data-points-in-r/) or another freeware/software/code capable of drawing Raincloud plots with jittering.

For example, please see Fig.2 at Bommakanti et al. (2023) or Appendix B provided by Liu and Liu (2024)

Bommakanti et al. Comparative Transcriptomic Analysis of Archival Human Vestibular Schwannoma Tissue from Patients with and without Tinnitus. J Clin Med. 2023;12(7):2642. https://doi.org/10.3390/jcm12072642

Liu, Liu. Aided Diagnosis Model Based on Deep Learning for Glioblastoma, Solitary Brain Metastases, and Primary Central Nervous System Lymphoma with Multi-Modal MRI. Biology (Basel). 2024;13(2):99. https://doi.org/10.3390/biology13020099

The visualized type of bands (SD, CI95 or other) must be indicated in the captions for the figures.

[4]

Since the authors have built own ML-based models and have assessed the third-party results using loistic regression analysis, the detailed step-by-step pipeline depicting the sequence of data processing/ evaluating procedures must be presented in a separate subsection of the “Materials and Methods” section for each procedure bot as a text and as a figure.

Please consider to use X or Y axis as a timeline axis, while drawing each pipeline (for “Predictors for the Occurrence of Preoperative Seizure”, for “Predictors for the Occurrence of Postoperative Seizure”, and for “Evaluation of the Modified STAMPE2 Score”.

[4a]

While building own logistic regression score, please identify full details of the model constructed.

In particular, please identify:

- whether factor-by-factor interactions have been accounted for in the model built.

- in which order independent variables have been entered into the analysis

etc

[5]

The details of the validation strategy used in this study to assess built regression-based models are missed in the text of the manuscript.

[5a]

In the relevant subsection of the “Materials and Methods” section, the authors must explain the validation strategies in full details.

Whether training, test, validation sets were formed, using the initial sample;

In which way these sets were formed (randomly, by blocks, etc)

Were the cross-validation or leave-one-out techniques applied?

[5b]

Due to the relatively large size of the sample (n>100), please consider to use cross-validation as a primary validation strategy used in this study for built models.

[5c]

If no validation procedures were applied, then the authors must indicate this serious limitation clearly in the “Limitation” subsection of the manuscript (within the “Discussion” section).

[5d]

Based on the validation strategy applied, the ROC curves must be depicted as an additional Figure with the corresponding uncertainty bands estimated via cross-validation strategy (or another relevant technique).

The code for such an extension of the analysis is available at

https://scikit-learn.org/1.1/auto_examples/model_selection/plot_roc_crossval.html

or at other sources.

[5e]

AUC performance indicators must be reported for multivariate models built, with AUC variances estimated via cross-validation or other technique.    

[6]

The data presented in the Table 5 must be analysed in a proper way.

[6a]

Initially, the estimation of proportions for the modified STAMPE2 score in patients with postoperative seizures must be done for all Table 5’s cells simultaneously via a RxC matrix strategy, not for individual STAMPE2 scores alone.

The task might be completed using online freeware “Epitools” (https://epitools.ausvet.com.au/chisq), which might depict CI95 intervals for cell numbers, or another statistical software/freeware capable to complete such a task.

In case of a proper analysis, the authors will receive the following values Chi-square (6)= 19.35, p=0.0036

[6b]

Please report the raw results of the RxC-estimated proportions and relevant CI95 intervals in the Table 5.

[6c]

In the text, please identify the STAMPE scores, which CI95s do not include the 0.5 value.

[6d]

Please consider to perform additional comparisons, unifying successive STAMPE2 scores in various ways (for example, 0-3 scores vs 4-6 scores, as well as 0-2 scores vs 3-6 scores). These comparisons might be reported as an additional primary or supplementary tables.

For such comparisons, please report relevant Chi-squares.

[6e]

Based on the gained results, please discuss these results extensively in the relevant subsection of the “discussion” section.

In particular, please discuss the following issue:

- The identified lesser threshold (3+ or 4+) for the modified STAMPE2 scores that makes possible an identification of patients at high risk for postoperative seizures in the analysed cohort, suggesting the modified STAMPE scores as a sensitive yet too understudied instrument to guide anticonvulsant treatment.

[6f]

Please avoid doing mechanistic statements like [LL34-35] and identify scientifically sound interpretations of the results gained.

[7]

The statement “Of the six factors included in the modified STAMPE2 score, only sensorimotor deficit appeared to be an independent predictor for postoperative seizures in our study population” is based on a simple regression analysis with unidentified procedures accounting for factor-to-factor interactions and variability.

Such regression analysis is a possible yet too suboptimal and rough tool to identify predictive variables for the occurrence of seizures.

[7a]

To ensure the sound analysis of the predictors, the authors must test an impact of individual factors on resulting modified STAMPE2 score values via relevant multivariate analyses.

As an entry point for such procedures, please consider to perform the traditional correlation-based PCA on the data and to report the results of PCA.

[7b]

Using PCA-based projections of cases on factor planes, the authors must initially identify the outlier cases that might affect the regression analysis.

[7c]

After excluding outliers, the authors must assess the number of factors predicting the variance in the dataset, using a scree-plot.

[7c]

Omitting outlier cases, it is possible to identify correlated variables underlying factors identified.

[7d]

Other multivariate techniques capable of assessing factor-to-factor interactions also might be used.

Minor issues

[8]

The Table 2 is duplicated. Please remove the excessive copy.

Please also note that the Table 2 must be remained within the primary manuscript and not moved to the Supplement after completing the point [3b] of this review.

[9]

[LL113-114] “Since postoperative EEG is not routinely performed at Department of Neurosurgery at Leipzig University Hospital, the score could not be applied to our cohort.” - could not be applied directly? Please clarify

[10]

The “Introduction” section should describe the story on STAMPE2 scores and modified STAMPE2 scores.

Comments on the Quality of English Language

Text and language issues

A number of issues must be resolved.

[11]

Some references were missed in the manuscript.

Please improve.

[L119] “Error! Reference source not found.”

[L165] “Error! Reference source not found. ”

[L184] “Error! Reference source not found. ”

[L207] “Error! Reference source not found. ” - Twice

[L223] “Error! Reference source not found. ”

[12]

Fuzzy words and wordings must not be used.

Some examples

[L76] “adequate documentation in medical records” – Please clarify, what is “adequate “.

[LL9-10] “leading to increased morbidity and mortality” of whom? – please clarify

[L39] “Accounting for almost forty percent” of what? – please clarify

[L110] “recraniotomy” – ? “re-craniotomy”

etc

[12]

Please omit redundant intros

As a single example

[L11] “For this purpose, our study aims to” –

[13]

Some sentences are bulky and must be simplified.

Examples

[LL42-45] “In this regard, …” – Simplify the sentence.

[LL47-50] “In addition, more…” – Simplify the sentence.

[LL83-85] “Since no anticonvulsant prophylaxis..” – Please improve word order in the sentence

[14]

Pronouns must be used in a clear way.

As a single example

[L41] “its benign origin” – Please clarify “its”.
